# Using Kolmogorov Entropy to Verify the Description Completeness of Traffic Dynamics of Highly Autonomous Driving

**Gabor Kiss** [1,*] and **Peter Bakucz** [2]

[1] Department of Computer Science, J. Selye University, 945 01 Komarno, Slovakia
[2] Institute of Safety Science and Cybersecurity, Obuda University, 1034 Budapest, Hungary; bakucz.peterl@bgk.uni-obuda.hu
[*] Correspondence: kissga@ujs.sk

**Abstract:** In this paper, we outline the analysis of a fully provable traffic system based on the Kolmogorov entropy. The completeness of the traffic node dynamics is realized in the form of a nonlinear dynamical model of the participating transport objects. The goal of this study is to determine the completeness of transport nodes based on the Kolmogorov entropy of the traffic trajectories of a node with an unspecified number of actors, like cars and pedestrians. The completeness of a highly autonomous driving detection system describing a traffic node could be realized if the entropy-based error-doubling time of the trajectories of the Euler–Lagrange equation interpreted at the transport junction is less than 1.3.

**Keywords:** system completeness; highly autonomous driving; Kolmogorov entropy

## 1. Introduction

Self-driving vehicles are a hot topic these days, but the idea is nothing new. Radio-controlled cars have existed since as early as 1925 [1]. In 1958, the Chrysler Imperial was capable of maintaining speed, similar to the way that cruise control works today [2]. In 1995, a Mercedes was equipped with an almost fully self-driving capability and covered 2000 km, although there was no room for anyone but the driver because of all of the electronic equipment [3].

To bring them into our everyday lives today, both new, more accurate sensors [4] and new research in artificial intelligence [5] are needed. Some research has focused on autonomous cruise control for remotely piloted ships [6] and decision-making related to maritime self-driving navigation [7] or autonomous underwater vehicles [8], but in this paper, we focus on road transport, which is a much more challenging area for researchers due to the complexity of traffic situations [9,10]. More and more researchers are taking up research in this complex area.

There is research that looks at how comfort and trust ratings are influenced by certain features of the fully automated, real-world, and shared vehicle experience [11]. There are a number of lines of research that are exploring ways to support autonomous simulations with better-quality map information [12]. There is research investigating fast and reliable V2X communication between self-driving vehicles to implement a collective detection service for traffic situations [13]. An important direction of the current research is the recognition of road conditions for autonomous driving using LiDAR [14] and exploring the links between sunny and adverse weather conditions for autonomous driving and driving simulation [15].

Research that is more closely related to machine learning, such as autonomous race driving with model predictive control [16] or cyber treats [17] and stability analysis for autonomous vehicle navigation [18], is also very important too.

Despite extensive research in the area of self-driving vehicles, we are still far from having fully self-driving cars (SAE level 5) on the roads.

One reason for this is the difficulty of applying automotive safety standards to autonomous driving systems [19,20].

So far, very few projects have addressed the issue of completeness in the analysis of deep learning systems for sensing in highly autonomous vehicles [21,22]. The main reason for this is that the complexity required to determine the completeness of the system dynamics of a given traffic node could not be provided by conventional methods, and this requires an extension of the operations [23–25].

The usability of highly and fully automated vehicles on the road has been one of the most debated issues since the first demonstrations of this technology were made. To answer this question, the following three basic dimensions need to be considered:

(1)    The legal/regulatory dimension.
(2)    The technical dimension (system design).
(3)    The reasoning dimension (proof).

The first step is to define the scope and context of the automated driving system. The basic function of the product is to transport a person or payload from point A to point B. For a commercial product, the system must also perform other functions. These will not be discussed further here.

The central issue for approval is whether the system meets all the requirements, in particular, those of the legislator and those of society, such as ethical requirements.

The operational domain of the AD system is called the operational design domain, ODD (SAE J3016). The ODD includes, among others, the geographical operational domain or the weather conditions, i.e., in general, the environment in which the AD system can be operated, according to the specification.

To gain an understanding of how an AD system works and functions, it is necessary to model the system. In particular, a comparison with the human driver is useful, on the one hand, to learn from the human information processing chain and to imitate or improve suitable elements, and on the other hand, to be able to argue the suitability of the system for public road traffic compared to a human driver to an approval authority.

Accident statistics show that accidents are caused by typical driving errors, the majority of which can be divided into three categories:

1.    Errors of perception (as a rule, a road user was not seen).
2.    Errors of judgment (the prediction of the situation and the behavior of others or one's own movement does not correspond to reality).
3.    Manual driving errors (the vehicle is not operated adequately).

Extensive technical systems have been developed in the past to compensate for these driving errors, particularly for type 3. These include ESP, ABS, emergency brake assist systems, etc. For the design of a technical system to solve the driving tasks of a human driver, this results in a sequence of steps that include the following:

1.    Object/environment perception.
2.    Scene formation.
3.    Prediction.
4.    Generation and evaluation of alternative actions.
5.    Action determination.
6.    Conversion of action into a movement.
7.    Re-evaluation of the influence of one's own behavior on the environment.

The approval of an AD system for safe and, at the same time, economically attractive use in public road traffic must satisfy the following several aspects:

1.    The system must represent a real benefit or safety gain, according to the state of the art. For RB, this must be achieved in the best possible way in accordance with RB/GF182.
2.    The system must be approved by the authorities. To this end, there must be comprehensible and officially recognized evidence and reasoning that justifies the safety and

regulatory compliance of the system and withstands a legal review of the engineering duty of care at all times.

3.   The system must be performant. The system must enable the transportation of persons or goods in road traffic in such a way that the legitimate expectations of users and operators are met.

The system must be able to move in road traffic without presenting an unacceptably low benefit or even an unreasonable obstacle for users and other road users.

To achieve the legal and regulatory requirements, these are analyzed and systematically included as mandatory requirements in the specification of the system at the beginning of the design phase. The approval regulations that are valid at the time are essential for official approval. Their compliance must be demonstrated through suitable activities. As things stand today, a sufficiently positive risk–benefit balance and proof of due diligence in the design and development of the system will be key points.

In general, the duty of care includes consideration of the state of the art in science and technology, consideration of the safety generally expected by the public, consideration of reasonably foreseeable misuse, conformity of the product, and compliance with or exceeding relevant norms and standards. For a pure black-box verification of this rate on the basis of endurance driving, driving performance in the range of $6 \times 10^7$ h would be required.

System probability and completeness are necessary for release processes. Therefore, in this article, we introduce a fully provable and complete dynamical system for a traffic node based on the Euler–Lagrange equations. A main prerequisite for the Euler–Lagrange equations for AD systems is proof of a sufficiently low risk in relation to the limited release. The fixed threshold value of $5 \times 10^{-8}$ h$^{-1}$ for the occurrence of an accident with the damage class S1–3 serves as the starting point.

The task is therefore to develop a method, approximating the Euler–Lagrange system or procedure, with which the residual risk that the occurrence rate does not meet the required verification target can be estimated on the basis of significantly fewer endurance driving hours.

One way to reduce the required endurance driving hours without losing confidence in the mathematical verification of the targets is to use a higher information density than the counting of accident damage events and to combine this with valid, justifiable models of the system behavior. In addition, further qualitative, expert-based methods are used to support the required assumptions.

The concept of entropy is used to approximate the Euler–Lagrange equations. And to evaluate the entropy concept in embedded systems in real time, we introduce Kolmogorov entropy.

The utilization of a higher information density can be achieved, for example, by measuring the internal interfaces in the signal processing chain and the definition of criticality metrics in combination with statistical extrapolation methods. A corresponding procedure has already been successfully developed for a reduction in the scope of continuous runs for the validation of AEB functions. Furthermore, there is the possibility of information enrichment and system modeling within the framework of SiL simulations.

The addition of models (e.g., in the SiL simulation, but also implicitly in criticality metrics) introduces assumptions whose validity can usually be justified but not proven. For this reason, several options for residual risk estimation are described below, which are based on different principles and checked against each other for plausibility.

After the basics of Euler–Lagrange equations and Kolmogorov entropy, we present the completeness determination. Finally, we present our first results and conclusions.

## 2. Methods

In highly autonomous driving engineering practice, it is often necessary to create a system that is provable. In addition to provability, the completeness of the system is also important and should be treated together in practice.

Generally, a provable system means that at all times of operation, it can be seen that the system components are working as intended.

Provability means that, for a traffic node (with actors like cars and pedestrians), the following are analyzed and quantified using the Euler–Lagrange system:

- The geometry;
- The traffic dynamics;
- The boundaries;
- The mathematical, algorithmic object detection of the node.

System provability and completeness are necessary for system release processes.

In this article, we introduce a method that defines the provable and complete descriptive dynamical system for traffic nodes based on the thermodynamics of the Euler–Lagrange system of a traffic node to allow for a full proof of the completeness of the descriptive dynamics (detection software). Generally, research usually focuses on the proof of correctness of a traffic node with environments, dynamics, and topology for a road junction (see Figure 1) to prove the completeness and correctness of the description, whereby they try to solve the analytical solution of Euler–Lagrange's differential equation, taking into account the geometrical, topological, physical, and dynamic parameters. The final task is to prove the completeness of the transport node, i.e., that the dynamics of the transport node can be described with the existing toolset (algorithms, hardware, and software). In this case, the blue vehicle turns left in a large curve, while the other vehicles want to continue straight.

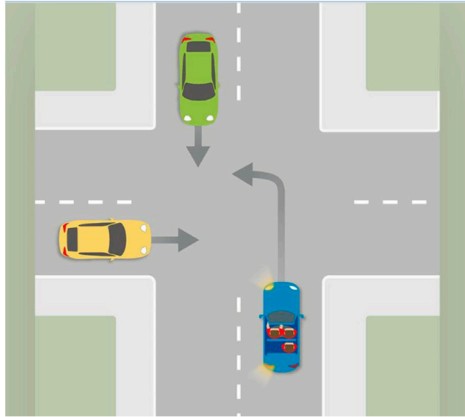

**Figure 1.** Basic environment of the problem.

Our ultimate goal could be to give a formalism on how to simply prove completeness and correctness.

But the greatest problem in proving the completeness of the transport node is that the resulting tool is not embedding compatible and the model used is linear, although measurements and experience show that the dynamics of the traffic node are nonlinear (not predictable). It is probably also known from our own experience that the behavior of traffic participants is not always reducible to ground rules and is sometimes chaotic.

There is also a need to create methods that occur in real time, which is usually desirable in most applications where computational time is critical. Consequently, in the embedded microcontroller environment, there is a need for a more efficient and accurate method to solve the completeness problem in the traffic node.

For the embedded microcontroller applications in the chassis control engineering literature, the Euler–Lagrange method combined with the globally converged Newton–Raphson method is well suited for this kind of problem for completeness and understanding traffic node dynamics with different sensor sets, actors, and boundaries. The advantages of this method lie in its fast convergence in the neighborhood of the optimal solution. The solutions can also be obtained with extremely high precision.

However, this method has two major disadvantages. First, it may converge to a <u>local</u> optimum → it requires good initial guesses that lie within the domain of convergence. This requirement is especially difficult because it involves the use of Lagrange multipliers, whose physical meaning is nonintuitive.

In particular, we present here a new method (beyond the Euler–Lagrangian) based on Kolmogorov entropy, i.e., the completeness of a node is not a partial differential equation description of the energy of the transport participants and the initial and boundary conditions of the system, but the information content of the renderer is analyzed in a real-time embedded environment.

### 2.1. The Euler–Lagrange Method

The basis of the Euler–Lagrange method, according to which the completeness of the system, is represented by the solvability of the system of partial differential equations [26] for a given traffic node, as we mentioned, since the realization of solvability is sometimes difficult, the study of Kolmogorov entropy is a prerequisite for the performance of the analytical solution.

We emphasize, which is the most important innovation of this article, that solving Euler–Lagrange equations in real time in embedded environments is not possible due to the energy requirements. Hence, we use the entropy concept for approximation. And to evaluate the entropy notion in embedded systems in real time, we introduce Kolmogorov entropy, as it is the most obvious procedure to fully cover the logarithm function of the classical Shannon definition.

In the Euler–Lagrange system, we approximate the dynamics of the traffic node with the variational principles, which apparently had a large impact on vehicle dynamics, approaching the various chapters on a single base. In many fields of autonomous driving, the basic equation of a physical and open-world principle can be discussed with "elegant simplicity" as the consequence of an extremal principle. However, several authors have already pointed out that this program cannot be performed on an "absolute general" basis [1–4]. Even in the most completely developed classical vehicle mechanics, all phenomena cannot be described by equations of motion derived from the Hamilton principle.

Although, according to their opinion, many variational principles can be created to a differential equation, creating variational principles is not a useless activity from our point of view, since not even vehicle mechanics easily leads to the solving of Newtonian equations and the solution, but using other principles is more effective (e.g., the analysis of impacts).

In the Euler–Lagrange system, the derivation of usual autonomous driving equations for release issues in an open-world context from Lagrange's principle will be addressed. The usual solution to autonomous driving problems involves the balance equations of mass, momentum, and angular momentum, in addition to the open-world environmental equation, an equation system that considers the initial and boundary conditions. It is a remarkable fact that the balance equation of kinetic and potential energy, which are the only mechanical energy forms in autonomous driving nomenclatures, is not necessary to the solution. Ziegler [27] also calls our attention to the significant differences between the momentum equations and the energy equations:

There is an essential difference between the momentum theorem demand and the energy theorem:

In the momentum theorems, the internal forces are absent;
In the energy theorem, they play an essential role (consideration of internal forces can be important, for example, to take into account hardware elements in the autonomous driving).

However, it is found that the derivation of formalism generally does not require kinetic and potential energy, and the number of elements, only the Lagrange density, but the density in the final equation, can be applied to both thermodynamic and kinetic and potential for the Schrödinger equation, i.e., systems used in self-driving automation.

It is very important that the trajectories represent "what happened at the node", and that this could be characterized as the descriptive dynamics of the traffic node. The descriptive means that we want to test software for object detection in traffic dynamics.

In highly automated driving engineering practice, it is often necessary to create a system that is fully demonstrable or provable. In addition to provability, the completeness of the system is also important.

In general, the duty of care includes consideration of the state of the art in science and technology, consideration of the safety generally expected by the public, consideration of reasonably foreseeable misuse, conformity of the product, and compliance with or exceeding relevant norms and standards. $\rightarrow$ The system provability and completeness are necessary for release processes. Therefore, in this article, we introduce a fully provable and complete dynamical system for a traffic node based on the Euler–Lagrange equations.

A main prerequisite for the Euler–Lagrange equations for AD systems is proof of a sufficiently low risk in relation to the limited release. The fixed threshold value of $5 \times 10^{-8}$ h$^{-1}$ for the occurrence of an accident with the damage class S1–3 serves as the starting point.

For a pure black-box verification of this rate on the basis of endurance driving, driving performance in the range of $6 \times 10^7$ h would be required.

The task is therefore to develop a method, approximating the Euler–Lagrange system or procedure, with which the residual risk that the occurrence rate does not meet the required verification target can be estimated on the basis of significantly fewer endurance driving hours.

One way to reduce the required endurance driving hours without losing confidence in the statistical verification of the targets is to use a higher information density than the counting of accident damage events and to combine this with valid, justifiable models of the system behavior. In addition, further qualitative, expert-based methods are used to support the required assumptions. The utilization of a higher information density can be achieved, for example, by measuring internal interfaces in the signal processing chain and the definition of criticality metrics (see Section 3.2) in combination with statistical extrapolation methods. A corresponding procedure has already been successfully developed for a reduction in the scope of continuous runs for the validation of AEB functions. Furthermore, there is the possibility of information enrichment and system modeling within the framework of SiL simulations. The addition of models (e.g., in the SiL simulation but also implicitly in criticality metrics) introduces assumptions whose validity can usually be justified but not proven. For this reason, several options for residual risk estimation are described below, which are based on different principles and checked against each other for plausibility.

Fully provable means, for a traffic node, that the geometry, the traffic dynamics, the boundaries, and the mathematical, algorithmic detection of the node are analyzed and quantified using Lagrangian densities. The dynamics of a transport node are determined by discretizing the node and recording the movements of each transport participant in a "state–next state" system.

For product owners in software engineering and system engineering, the procedure may be important to design a system where completeness and provability are ensured from the outset.

As part of the highly autonomous vehicle project at Óbuda University, we analyze the safety and reliability of the system in a separate project based on the Euler–Lagrange system. The goals of this project were as follows:

1. To determine completeness based on the Euler–Lagrange system.
2. To create a topological model for a real traffic node.
3. To collect traffic data on the intersection.
4. To analyze the completeness of the traffic node using the Euler–Lagrange system,

Determining the completeness of a system describing the descriptive dynamics of the detectability of a traffic node based on the entropy analysis of the solvability of Euler–Lagrange equations is an important application area.

The main task of this article is to define a method to create and design a mobility solution system that is completely (releasable in the open-world context) based on the dynamical Lagrangians, boundary conditions, and the experience of the detection knowledge.

The advantages of the method are as follows:

1.  The analysis of entropy is equivalent to examining the solvability of the Euler–Lagrange equation, which is a mainstage of the completeness analysis of traffic node descriptive dynamics (object detection software).
2.  The method is valid for an unspecified number of participants.
3.  Analyzing entropy is much simpler than other methods for examining the analytical solutions of the Euler–Lagrange equation.
4.  The method helps test a system that is consistent, complete, and releasable.
5.  The previous know-how, which means detection, perception, classification, etc., can be integrated into the entropy analysis system.
6.  The system can be easily expanded (more cars, pedestrians, boundaries, sensor sets for the detection, algorithmic experience, and different numerical methods (neural nets and moving averages)).
7.  The procedure can be used not only to check the completeness of the systems but also in the design, as we can change and prescribe the condition of the elements involved in the traffic situation until the proportion of attractor areas is negligible and the system has a dominant attractor.
8.  When comparing different traffic nodes, analyzing via entropy is simple and has advantages in an embedded real-time environment.
9.  The method can also be applied to embedded real-time systems.

The Euler–Lagrange Equations

The intent of this section is to give a brief look at the idea of Euler–Lagrange problems and to give enough information to allow us to conduct some basic partial differential equations in the next chapter. Now, with that out of the way, the first thing that we need to do is define just what we mean by a Euler–Lagrange problem. With the initial value problems, we had a differential equation, and we specified the value of the solution and an appropriate number of derivatives at the same point (collectively called initial conditions). For instance, for a second-order differential equation, the initial conditions are (1) as follows:

$$y(t_0) = y_0 \quad y\prime(t_0) = y\prime_0 \tag{1}$$

With Euler–Lagrange problems, we will have a differential equation and we will specify the function and/or derivatives at different points, which we will call Euler–Lagranges. For second-order differential equations, which we will be looking at pretty much exclusively here, any of the following can and will be used for the boundary conditions (2)–(5):

$$y(x_0) = y_0 \quad y(x_1) = y_1 \tag{2}$$

$$y\prime(x_0) = y_0 \quad y\prime(x_1) = y_1 \tag{3}$$

$$y\prime(x_0) = y_0 \quad y(x_1) = y_1 \tag{4}$$

$$y(x_0) = y_0 \quad y\prime(x_1) = y_1 \tag{5}$$

As mentioned above, we will be looking pretty much exclusively at Euler–Lagrange second-order differential equations. We will also be restricting ourselves down to linear differential equations. So, for the purposes of our discussion here, we will be looking almost exclusively at differential equations in the form (6),

$$y'' + p(x)y' + q(x)y = g(x) \tag{6}$$

Along with one of the sets of boundary conditions given in 1–4. We will, on occasion, look at some different boundary conditions, but the Euler–Lagrange differential equation will always be one that can be written in this form. As we will soon see, much of what we know about initial value problems will not hold here. We can, of course, solve 5, provided the coefficients are constant and for a few cases in which they are not. None of that will change. The changes (and perhaps the problems) arise when we move from initial conditions to boundary conditions. A differential equation is homogeneous if $g(x) = 0$ for all $x$. Here, we will say that a Euler–Lagrange problem is homogeneous if, in addition to $g(x) = 0$, we also have $y_0 = 0$ and $y_1 = 0$ (regardless of the boundary conditions that we use). If any of these are not zero, we will call the Euler–Lagrange nonhomogeneous. It is important to now remember that when we say homogeneous (or nonhomogeneous), we are saying something not only about the differential equation itself but also about the boundary conditions as well. The change that we are going to see here comes when we go to solve the Euler–Lagrange problem. When solving linear initial value problems, a unique solution will be guaranteed under very mild conditions. We only looked at this idea for first-order IVPs, but the idea does extend to higher-order IVPs. With Euler–Lagrange problems, we will often either have no solution or many infinite solutions, even for very nice differential equations that would yield a unique solution if we had the initial conditions instead of boundary conditions. Before we get into solving some of these, let us next address the question of why we are even talking about them in the first place. As we will see in the next chapter, in the process of solving some partial differential equations, we will run into Euler–Lagrange problems that will need to be solved as well. In fact, a large part of the solution process will have to deal with the solution to the BVP. In these cases, the boundary conditions will represent things like the temperature at either end of a bar or the heat flow into/out of either end of a bar. Or maybe they will represent the location of the ends of a vibrating string. So, for the boundary conditions, there will be conditions on the boundary of some processes. So, with some of the basic stuff out of the way, let us find some solutions to a few Euler–Lagrange problems. It must be noted as well that there really is not anything new here yet. We know how to solve the differential equation, and we know how to find the constants by applying the conditions. The only difference is that here we will be applying boundary conditions instead of initial conditions.

One excellent way to 'analytically' solve the Euler–Lagrange differential equation in real time in an embedded environment is the entropy method. The essence of this method is to represent the completeness or incompleteness of the system by the changes in the value of an indicator (in this case, Kolmogorov entropy) interpreted in the system of equations [28,29].

It might look odd that a powerful concept. such as entropy, whose importance is extended but not limited to mechanics, thermodynamics, information theory, and geo-statistics, has not found an essential role in other disciplines, like, for instance, the systems theory field. Only a few attempts have been made [30,31] to introduce entropy-based tools into control theory. In this paper, an entropy-based approach, which is derived from the theory of the thermodynamics of curves, is suggested and adapted for systems theory applications [32]. A new entropic indicator is introduced with interesting properties. It is different from other known algorithms used to classify nonlinear systems or attracting sets, like Lyapunov exponents or dimension-like concepts [33]. The classification of nonlinear systems is somehow a tough subject, and there is not yet a systematic approach to tackle it, at least to the authors' knowledge. Some attempts can be found where invariant moments of distribution of particles undergoing a Hamiltonian system are investigated [34,35], and where the classification of nonlinear stochastic systems is performed using input–output measurements. In this paper, a new indicator is proposed that has the property of providing a unitary value whenever applied to linear systems. Moreover, the proposed indicator does not depend on stability issues, and it is generally able to describe the disorder or irregularity in the evolution of a dynamic system. By irregularity, this means the distance from an ordered sequence of points along a line. The proposed indicator can be used within

any dynamic system; differences arise when the steady state is an equilibrium point, the attracting set is a periodic sequence, or it presents chaotic behavior.

The next section of our article is dedicated to a review of the entropy of curves, and basic concepts from the thermodynamics of curves are recalled. We present how the entropy indicator has been developed within a system theory framework, and the most important properties of the proposed algorithm are proved.

### 2.2. Kolmogorov Entropy Method

As mentioned above, in this project, we want to formulate a complete description of the traffic situation in real time. To do this, while keeping the basics, it is necessary to revolutionize the elaboration mechanism to implement the system on the embedded microcontroller.

This is achieved by introducing Kolmogorov entropy. The analytical solution of the Euler–Lagrange system for a traffic node could be described (7)

$$\{Y_n\}, n = 1, \ldots, N \tag{7}$$

and can be unfolded in a multi-dimensional effective phase space using time delay coordinates ($t$ is a delay time (embedding)) (8).

$$Y_n = (Y_{n-(m-1)t}, \ldots, Y_n) \tag{8}$$

An important observation is that the Euler–Lagrange equation always has a $\{Y_n\}$ solution, but a distinction must be made between the solutions that belong to the analytical solutions and those that do not, i.e., in which case we can make statements for the completeness of the node.

If $\{Y_n\}$ is an analytical solution of a Euler–Lagrange dynamical system for a given traffic scenario, it can be shown under certain genericity conditions that the reconstructed point set is a one-to-one image of the original trajectory of the traffic node dynamical system, where completeness and provability are to be defined.

In our article, to test the comparability of the traffic node, we use Kolmogorov entropy [36], which is an important characteristic describing the degree of solvability of the Euler–Lagrange system [37]. The entropy gives the average rate of information loss about the position of the phase point on the attractor.

In our article, we are looking for the traffic node, which is completely provable and consistent if (9)

$$0 < K < \infty \tag{9}$$

Incidentally, our entropy method can also be used to characterize the existing traffic node state because of the following:

- $K$ is infinite in a random traffic node dynamics system -> the system is not complete.
- $K = 0$ in a chaotic traffic node dynamics system -> the system is not complete.

The Kolmogorov entropy $K_q$ can be defined as follows; -> Let $Y(t)$ and $t > 0$ be the trajectory of the Euler–Lagrange equation for a traffic node in an $n$-dimensional phase space sampled at discrete time intervals $\Delta t$. Let us divide the phase space into the $n$-dimensional hypercubes of side $r$. Let $P_{i1,i2,\cdots,iN}$ be the joint probability that the trajectory $Y(t)$ on the Euler–Lagrange equation subsequently visits cubes $i_1$, $i_2$, $\cdots$, and $i_N$ at times $t = \Delta t$, $t = 2\Delta t$, $\cdots$, and $N\Delta t$.

The Kolmogorov entropy Is then ($1''$)

$$K_q = -\lim_{r \to 0} \lim_{\Delta t \to 0} \lim_{N \to \infty} \frac{1}{N\Delta t} \frac{1}{q-1} ln \sum_{i_1 i_2 \ldots i_N} P^q_{i_1 i_2 \ldots i_N} \tag{10}$$

In our article, we use the $K_2$ form (11)

$$K_2 \sim \lim_{m\to\infty}\lim_{r\to 0} K_2^m(r), \tag{11}$$

where (12)

$$K_2^m = \frac{1}{k\Delta t} ln \frac{C^m(r)}{C^{m+k}(r)} \tag{12}$$

and $k$ is a sufficiently small integer number and $m$ is the embedding dimension. The saturation value of $K_2^m$ as m increases is regarded as $K_2$ and $C^m(r)$, and the correlations integral defined by (13)

$$C^m(i,r) = \frac{1}{M-1}\sum_{\substack{j=1 \\ i\neq j}}^{M} \theta\left(r - \|Y_i - Y_j\|\right) \tag{13}$$

where $M = N - (m-1)/\Delta t$ and $N$ is the number of data points in the Euler–Lagrange trajectory. $\theta(x)$ is a Heaviside step function defined by (14)

$$\theta(x) = \begin{Bmatrix} 1 \; for \; x \geq 0 \\ 0 \; for \; x < 0 \end{Bmatrix} \tag{14}$$

In the following, we show how we can use entropy to prove the completeness of a transport node.

## 3. Results

### 3.1. The Determining the Completeness

There is a certain degree of uncertainty in estimating the entropy from Euler–Lagrange trajectories. In our article, in order to reduce fluctuations and improve the statistics, the formula, $K_2^m$ is averaged over five values computed for different embedding dimensions. This yields (15)

$$K_2^m(r) = \frac{1}{L}\sum_{l=1}^{L} \frac{1}{2l\Delta t} ln \frac{C^{m-2l}(r)}{C^m(r)}, \;\; L = 5 \tag{15}$$

The dependence of $K_2^m$ on the embedding dimension $m$ is approximated by means of least-squares fit by the function (16)

$$f(x) = a + \frac{b}{x^c} \tag{16}$$

This function converges to $a$ for $c > 0$ and $x \to \infty$. Therefore, (17)

$$K_2^m(r) = K_2(r) + \frac{b}{m^c}, \;\; c > 0 \tag{17}$$

$b$ and $c$ are real parameters, $m$ is the embedding dimension, and $K_2(r)$ is the entropy, which depends on the selection of the scaling region. Parameters $a$, $b$, and $c$ can be determined from real traffic node dynamic experiments.

The entropy-based value of an error-doubling time is the most important parameter analyzing the solvability of the Euler–Lagrange system and the completeness of the traffic node -> the entropy-based error-doubling time is defined as follows (18):

$$T_2 = \frac{ln2}{K_2} < 1.3 \to complete \tag{18}$$

If the entropy-based error-doubling time of the Euler–Lagrange trajectory of a given traffic node with Lagrangians of the following is less than 1.3 → the system is complete and provable:

— The geometry.
— The traffic dynamics.
— The boundaries.
— The mathematical and algorithmic detection of the node.

In Figure 2, the flowchart of the method can be seen.

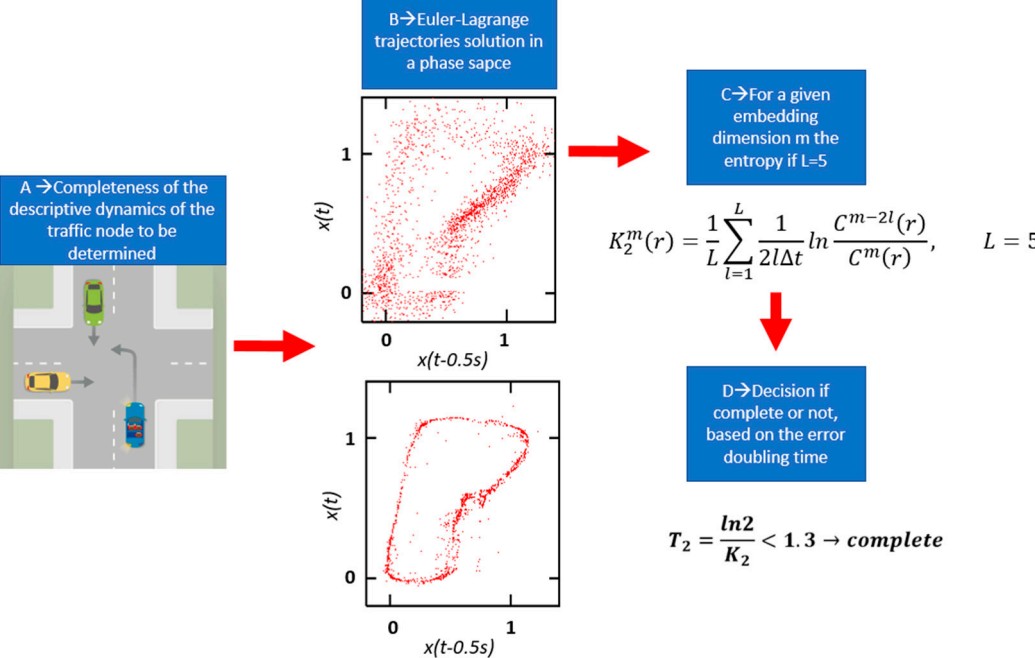

**Figure 2.** Flow chart of the system.

The steps are as follows:

A:    The completeness of the descriptive dynamics of the traffic node is to be determined.
B:    Euler–Lagrange trajectories solution in a phase space.
C:    For a given embedding dimension m the entropy if $L = 5$.
D:    Decision, if complete or not, based on the error doubling time.

### 3.2. Real Measurements

In the project on self-driving cars at the University of Óbuda, we captured a variety of convergence situations that provide a broad cross-section of the different traffic dynamics, the characterization of the completeness of which in the embedded environment and in real time is of utmost importance.

Table 1 presents a description of the experiments used to describe the completeness of transport systems, a description of the sensor set, a description of the location and time of the experiments, and a summary of the results.

Figure 3 shows the traffic scenarios recorded with a video camera and a corner radar set. The task is to determine the completeness of the description of the traffic situation dynamics using Kolmogorov entropy (Panasonic Automotive Systems Camera, Langen Germany).

Figure 4 shows the plots of the Kolmogorov entropy variation with respect to the six experiments performed. The system is considered complete, i.e., perfectly described by the sensors, if the entropy is at the upper (near 0) position. Outliers indicate that at a given moment, the system cannot give a clear answer about the traffic situation based on the video image. In this case, another sensor needs to be activated.

**Table 1.** The table characterizes and describes the experiments carried out. Here, we also present the results of the Kolmogorov entropy-based teleconditioning, i.e., whether the sensor set is sufficient to obtain the system's completeness in a given traffic situation.

| Experiment | Who | Sensorset | Time | Results of the Entropy Method |
|---|---|---|---|---|
| 1. Urban traffic | Budapest, Hungary | Panasonic Video Camera (Panasonic Automotive Systems Camera, Langen Germany), 4 HTU Corner Radar | June 2023 | The many outliers (incomplete) in urban traffic show that the entropy method is an excellent way to demonstrate the inadequacy of the sensor set. In this case, many more sensors would be needed. |
| 2. Urban Motorway | Budapest, Hungary | Panasonic Video Camera, 4 HTU Corner Radar | August 2023 | On the urban motorway, we can characterise the completeness of the system in terms of entropy in the same way as for urban traffic. |
| 3. Highway | M1 highway near Budaörs Hungary | Panasonic Video Camera, 4 HTU Corner Radar | July 2023 | Entropy-based completeness testing for motorways is of limited use because of all the outliers. |
| 4. Rural minor road | Füle, Near Polgárdi Hungary | Panasonic Video Camera, 4 HTU Corner Radar | June 2023 | On the rural minor road, the system's completeness can be fully guaranteed by the Panasonic video sensor set, as the entropy shows (no outliers) |
| 5. Montain Road | Gödöllő, Hungary | Panasonic Video Camera, 4 HTU Corner Radar | August 2023 | for mountain road at low speeds, the system completeness, based on Kolmogorov entropy, is adequate. |
| 6. Working on the Highway | M1 highway near Tata Hungary | Panasonic Video Camera, 4 HTU Corner Radar | June 2023 | the low speed and lack of complexity before working on the motorway makes the system complete, as shown by the entropy |

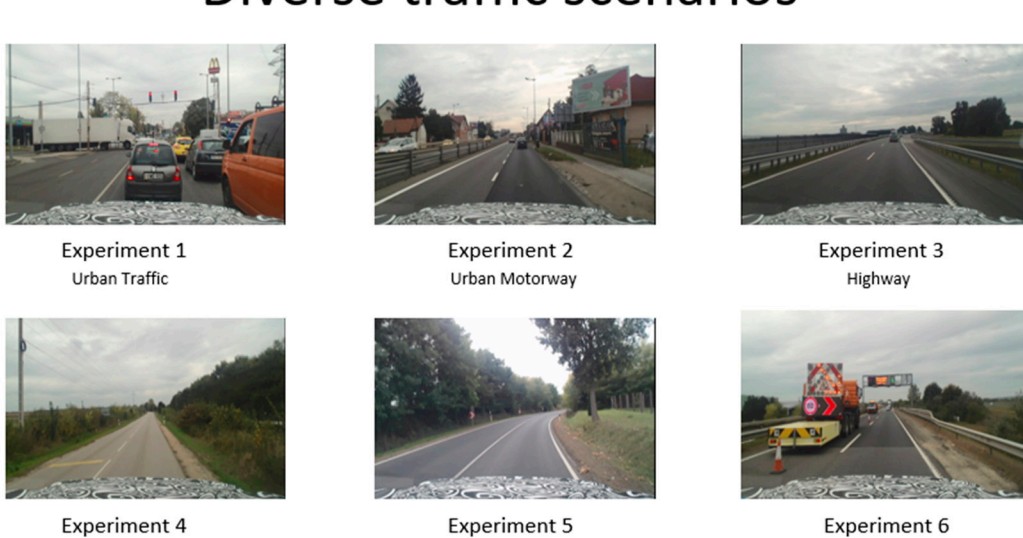

**Figure 3.** Diverse used traffic scenarios. The main issue is the completeness of the description of the traffic situation dynamics using Kolmogorov entropy.

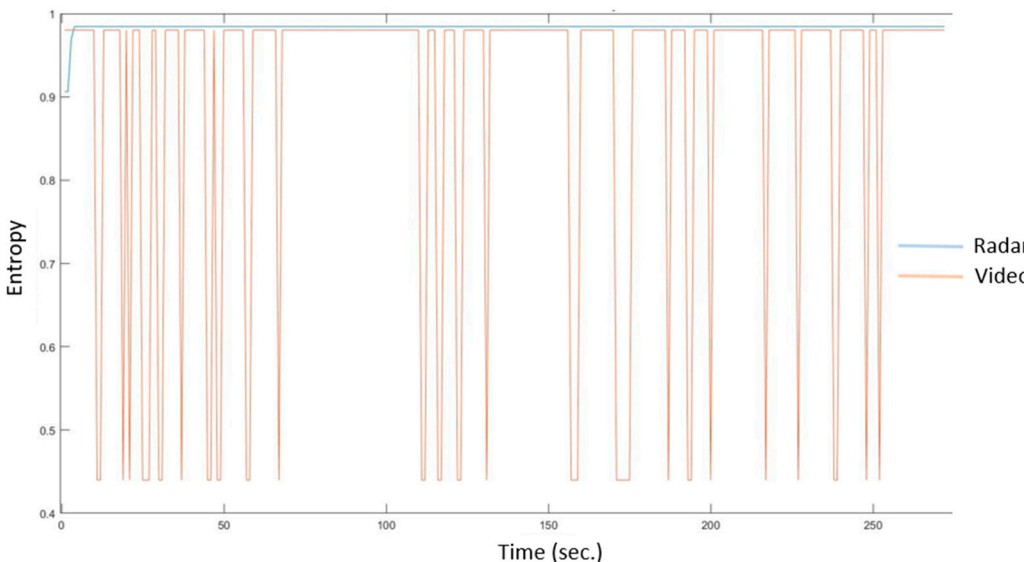

**Figure 4.** Kolmogorov entropy, Experiment 1. Urban Traffic.

The following figures show the Kolmogorov entropy of the traffic scenarios (Figures 4–9).

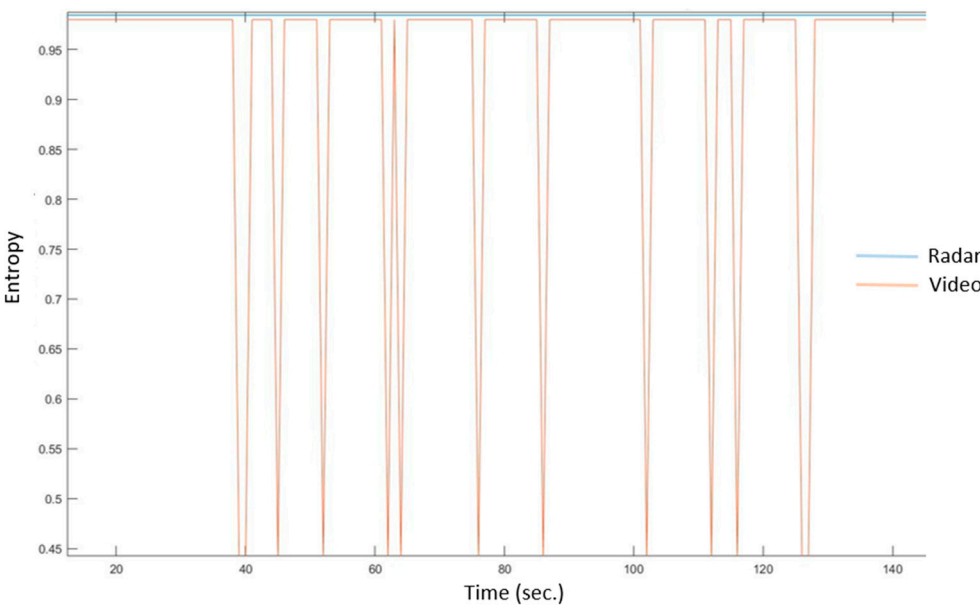

**Figure 5.** Kolmogorov entropy, Experiment 2. Urban Motorway.

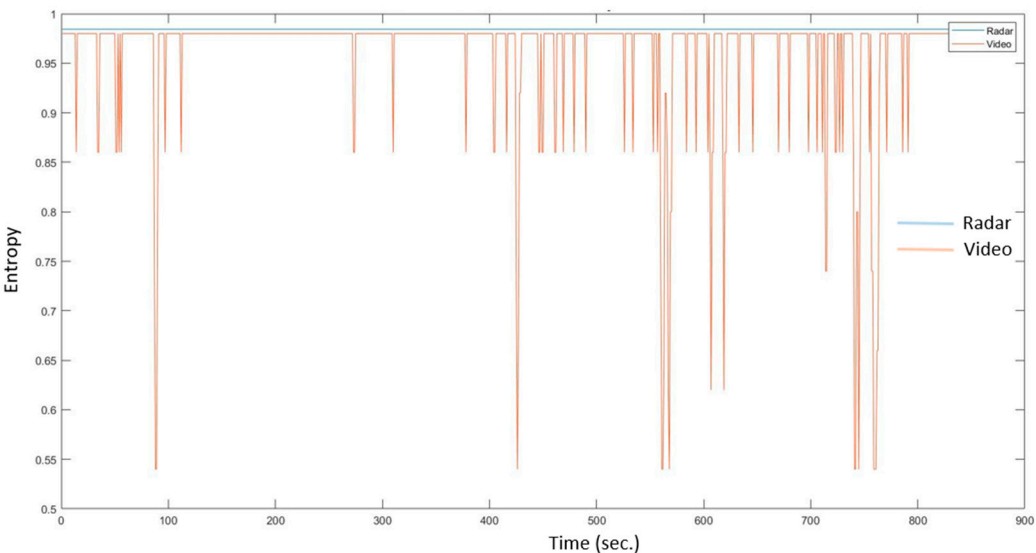

**Figure 6.** Kolmogorov entropy, Experiment 3. Highway.

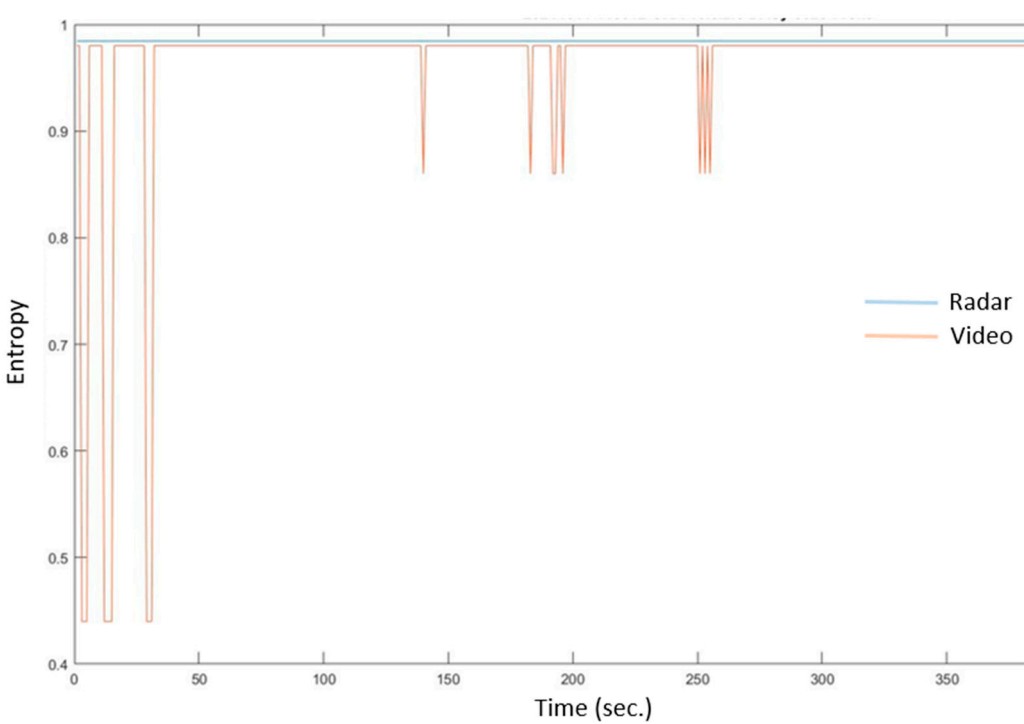

**Figure 7.** Kolmogorov entropy, Experiment 4. Rural Minor Road.

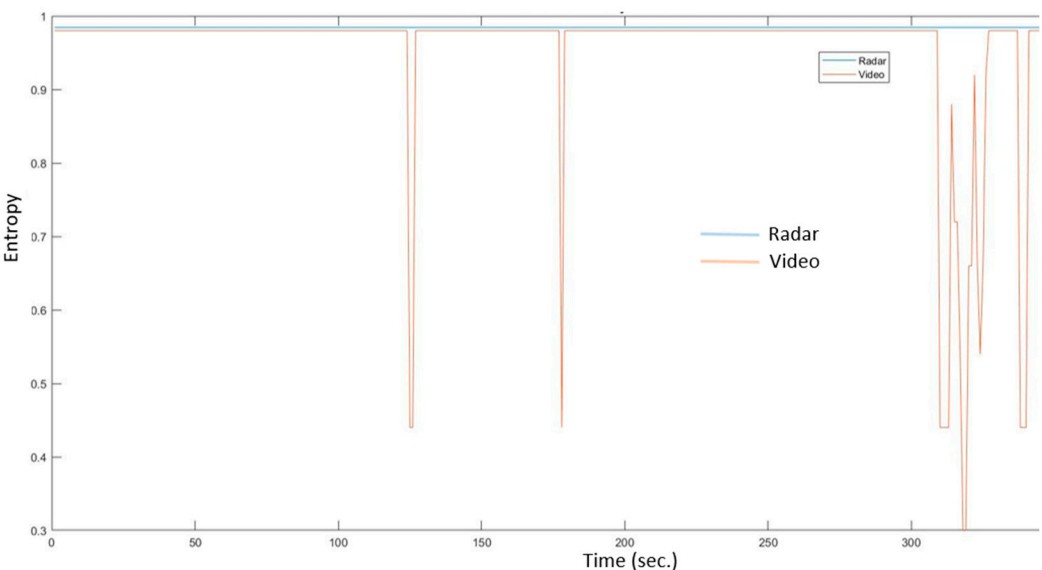

**Figure 8.** Kolmogorov entropy, Experiment 5. Mountain Road.

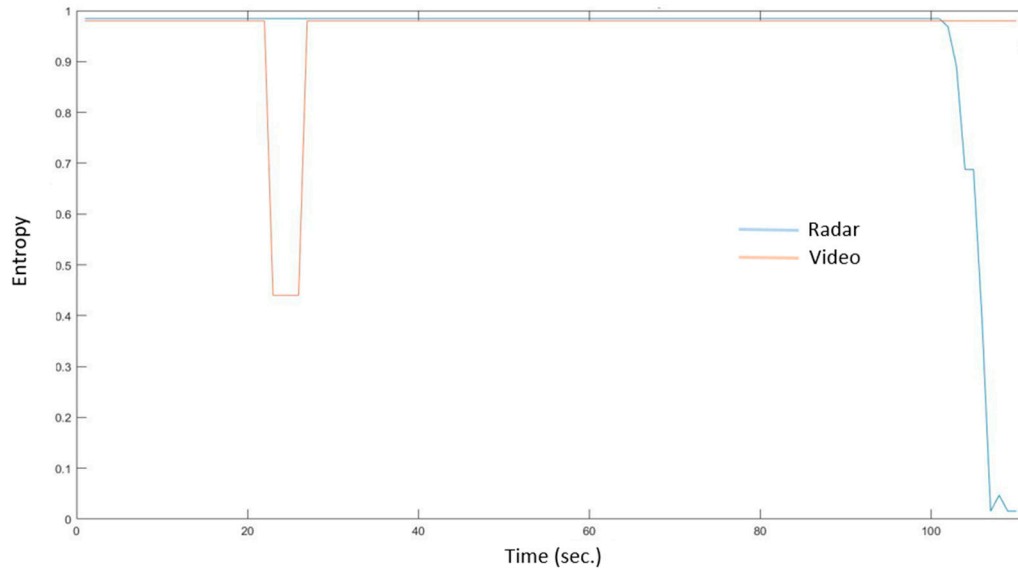

**Figure 9.** Kolmogorov entropy, Experiment 6. Working on the Highway.

The system is considered complete, i.e., perfectly described by the sensors, if the entropy is at the upper (near 0) position. Outliers indicate that at a given moment, the system cannot give a clear answer about the traffic situation based on the video image. In this case, another sensor needs to be activated.

The main problem that we have addressed in this paper is to develop a measure of the completeness and provability of a given AD system in real-time and under embedded conditions. For this purpose, we have used Euler–Lagrange equations as the mathematical basis, which cannot be solved in real time, and an embedded system without large energy consumption.

The Euler–Lagrange equations are approximated by Kolmogorov entropy. Figures 4–9 show time on the horizontal axis and entropy on the vertical axis. This entropy can be seen, for example, on the interior screen of the AD car, i.e., the L3–L4 driver can see the behavior of the system in real time.

What does the figure mean? The greater the outlier content of the Kolmogorov entropy, the more likely the system will remain chaotic and uncertain going forward, because the

sensors do not provide enough information (their entropy changes suddenly) to perform the perception, and so an additional sensor has to be switched on. The figures show a typical entropy pattern, as indicated in Table 1, when it is necessary to consider switching on an additional sensor, which of course, results in an increase in the energy demand.

## 4. Conclusions

The main task of the article was to define a method to test a system that is provable and complete (releasable in the open-world context) based on the Kolmogorov entropy analysis of the Euler–Lagrange equation, boundary conditions, and the experience of the detection knowledge.

The main issue of the article is as follows:

The completeness of the detection system describing a traffic node could be realized if the entropy-based error-doubling time of the trajectories of the Euler–Lagrange equation, interpreted at the transport junction, is less than 1.3. The error doubling is defined as follows (19):

$$T_2 = \frac{ln2}{K_2} < 1.3 \tag{19}$$

where $K_2$ is the Kolmogorov entropy of the trajectory. The analysis of the error doubling is equivalent to testing the solvability of the Euler–Lagrange system.

The advantages of the method are as follows:

1. It helps the designer build a system that is consistent, complete, and releasable.
2. Due to the application of the variation principle, the created system becomes extreme, i.e., we can find the minimum or maximum of all existing systems.
3. The previous know-how, which means detection, perception, classification, etc., can be integrated into the system.
4. The system can be easily expanded (more cars, boundaries, sensor sets for detection, and the complexity of the node can be changed), and then all we have to do is solve the partial differential equation to see whether the completeness or provability of the system exists. If not, what decisions need to be made to ensure completeness.

We have presented tests of extensive real traffic scenarios ((1) urban traffic, (2) urban highway, (3) freeway, (4) minor road, (5) mountain road, and (6) construction area on a highway) and found that the Kolmogorov entropy is an excellent tool to characterize the completeness of the traffic situation in real-time embedded environments. The outliers observed in Figure 4 show that in this case, the information content of the system is not sufficient, and another additional sensor needs to be activated.

The next steps in modeling are as follows:

1. Recording the traffic dynamics of the selected traffic node.
2. Constructing the Euler –Lagrange system and using it to calculate the completeness.
3. Creating a small-scale model and sensor set and performing traffic experiments.
4. Comparing and scaling the results.

**Author Contributions:** Conceptualization, G.K. and P.B.; methodology, P.B.; software, P.B.; validation, G.K. and P.B.; formal analysis, G.K.; investigation, P.B.; resources, P.B.; data curation, G.K.; writing—original draft preparation, P.B.; writing—review and editing, G.K.; visualization, P.B.; supervision, G.K. All authors have read and agreed to the published version of the manuscript.

**Funding:** This research received no external funding.

**Institutional Review Board Statement:** Not applicable.

**Informed Consent Statement:** Not applicable.

**Data Availability Statement:** Data is contained within the article.

**Conflicts of Interest:** The authors declare no conflict of interest.

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
