# Peer review of "Using Kolmogorov Entropy to Verify the Description Completeness of Traffic Dynamics of Highly Autonomous Driving"

_applsci, doi:10.3390/app14062261_

Round 1

Reviewer 1 Report

Comments and Suggestions for Authors

Generally article is good/ have some hints to above:

1. are all references are appropriate is there room to involve any other regarding specific problem of autonomy of vehicles such as are rules on the road in the complex traffic hubs;

2. authors mentioning in further research new sensors implementation (is there are any specific to be mentioned in article);

3. in introduction to be more specific why this article is enhanced comparing to others in same area of expertise?

Author Response

Dear Reviewer,

thank you for your hard work to check our paper!

Reviewer 2 Report

Comments and Suggestions for Authors

The pape needs editoral nad formal improvments.

For example "we now focus on this area and try to provide a solution to the problem, how exactly do you define 'problem'?

SOTA should be expanded.

Definitely more concrete results should be presented. The article in my opinion presents a general description of the idea with residual results. More concrete numerical results are needed perhaps in the form of tables or graphs.

Author Response

Dear Reviewer,

thank you for your hard word to check our paper,

We extended the research design, described the methods in detail , illustrated the results through examples, and detailed the conclusions. 

 We look forward to your positive review.

Reviewer 3 Report

Comments and Suggestions for Authors

The presented article addresses the critical issue of establishing a provable and complete system for highly autonomous driving, focusing on the dynamics of traffic nodes. By employing the Euler-Lagrange equation and introducing the innovative use of Kolmogorov entropy, the authors propose a comprehensive methodology for assessing the completeness of the descriptive dynamics of traffic nodes.

The article presents a systematic approach that assists designers in constructing systems that are not only consistent and complete but also release-ready. Leveraging the variation principle, the method optimally captures extreme values within the system, providing a versatile framework applicable to diverse scenarios. Notably, the integration of previous knowledge, including detection, perception, and classification, adds a valuable layer to the proposed system. The flexibility of the system, allowing easy expansion and modification, provides a pragmatic solution for real-world applications.

Validation and Verification: It is imperative to provide extensive validation and verification of the proposed methodology. Empirical evidence and comparative studies against existing approaches or real-world data would strengthen the credibility of the presented results.

Consider discussing potential challenges and practical considerations for implementing the proposed methodology in real-world scenarios. Addressing issues such as computational efficiency, scalability, and real-time applicability will enhance the practical relevance of the method.

Encourage further exploration of the proposed method in diverse and complex traffic scenarios. Investigate its robustness under various conditions and its adaptability to evolving technologies and traffic dynamics.

As highly autonomous driving involves interdisciplinary aspects, consider exploring collaborations with experts in related fields, such as control engineering and transportation engineering, to enrich the analysis and contribute to a holistic understanding.

In conclusion, the article offers a promising approach to assessing the completeness of traffic node dynamics in highly autonomous driving systems. Addressing the provided suggestions would contribute to the method's robustness and applicability, fostering advancements in the field of autonomous transportation systems.

Author Response

Dear Reviewer,

we extended the references and the research design, described the methods in detail , illustrated the results through examples, and detailed the conclusions. 

 We look forward to your positive review.

Reviewer 4 Report

Comments and Suggestions for Authors

The paper analyzes a very interesting and current topic, however, before publication, I suggest a major revision. I have read the manuscript and I have found its quality below standard when compared to the papers published in the journal. I suggest the following improvements to the work:

-        The first impression is that there has been little effort when it comes to providing context and motivation. For this reason, enhance your summary and introduction with a more detailed description of the existing problem you want to solve;

 -        The methodology is scarcely explained. Please explain in detail why you chose the Kolmogorov Entropy;

 -        Figure 1 should contain a short title, while its explanation should be in the paper;

 -        The paper contains many general formulas, which makes reading less interesting;

 -        The results are clear and concise, but their breadth needs to be improved. Also, there is a lack of discussion on how the results obtained would be compared with the results from other studies;

 -        The conclusion should be corrected and contain explicit conclusions obtained in this paper, then, limitations of the study, future directions of research, and examples of practical applications of the obtained results;

 -        More recent references are missing;

 -        Please cite scientific papers and not rely on newspaper articles.

Author Response

(The authors gave the same response as above.)

Round 2

Reviewer 4 Report

Comments and Suggestions for Authors

Dear authors,
You have improved your paper. However, please review again the guidelines provided to you in the previous round of review; I believe the paper needs further enhancement. Figure 4, which has been added, needs to be clearer. The same comment applies to its title, as was mentioned for Figure 1 in the previous round of review. Kindly mark the changes in the text in the next submission so that the reviewers can more efficiently evaluate the paper.

Author Response

Dear Reviewer,

we have tried to take into account in detail the suggestions for modification.
We trust that our research will be accepted for publication this time.
Thank you for your thorough work and comments.